# Cold Agglutinin Disease: A Distinct Clonal B-Cell Lymphoproliferative Disorder of the Bone Marrow

**Fina Climent [1,*], Joan Cid [2] and Anna Sureda [3]**

1    Department of Pathology, Hospital Universitari de Bellvitge, IDIBELL, L'Hospitalet de Llobregat, 08907 Barcelona, Spain
2    Apheresis & Cellular Therapy Unit, Department of Hemotherapy and Hemostasis, ICMHO, IDIBAPS, Hospital Clínic, University of Barcelona, 08036 Barcelona, Spain; jcid@clinic.cat
3    Department of Hematology, Institut Català d'Oncologia, IDIBELL, L'Hospitalet de Llobregat, Universitat de Barcelona, 08907 Barcelona, Spain; asureda@iconcologia.net
*    Correspondence: fcliment@bellvitgehospital.cat

**Abstract:** Cold agglutinin disease (CAD) is a distinct clinicopathologic entity characterized by clonal B-cell lymphoproliferative disorder in the bone marrow. B-cell gene mutations affect NF-KB as well as chromatin modification and remodeling pathways. Clonal immunoglobulins produced by B cells bind to red cells (RBCs) at cold temperatures causing RBC aggregation, complement cascade activation and cold-autoantibody autoimmune hemolytic anemia (cAIHA). The clinical picture shows cold-induced symptoms and cAIHA. Therapeutic options include "wait and watch", rituximab-based regimens, and complement-directed therapies. Steroids must not be used for treating CAD. New targeted therapies are possibly identified after recent molecular studies.

**Keywords:** autoimmune hemolytic anemia; cold agglutinin; B-cell lymphoproliferative disorder

## 1. History

As early as 1529, a case of anemia after the exposure to cold was described by Johannes Actuarius, although Dressler is generally credited with being the first person to give a clear description of an autoimmune hemolytic anemia (AIHA), probably paroxysmal cold hemoglobinuria (PCH), in 1854 [1]. Nowadays, it is surprising to us that PCH was the first described AIHA in the latter part of the nineteenth century, given that PCH is the least common type of AIHA. However, its early recognition is due to the fact that hemoglobinuria is a striking symptom and it is also true that PCH was much more common in the past, due its association with late-stage syphilis or congenital syphilis. In the early years of the twentieth century, a distinction of congenital and acquired forms of hemolytic anemias was well defined, though the underlying mechanism was not well understood [2]. The description of the antiglobulin test by Coombs, Mourant, and Race in 1945 was a sight for sore eyes [3]. The direct antiglobulin test (DAT) not only distinguished between the familial and acquired forms of hemolytic anemia, but also demonstrated a difference in their etiology. Finally, in 1951, Young et al. were the first to coin the term AIHA [4]. Those authors theorized that the production of an autoantibody leads to autoimmunization.

Cold agglutinins (CA) were recognized in 1903 by Landsteiner in animal blood and in human blood by Mino in 1924. Their role in human disease was not recognized until 1943 when Stats and Wasserman published a review in which they stated that CAs could be innocuous in the great majority of cases, although in some cases cold hemagglutination was of pathogenetic significance [2]. In 1953, Schubothe introduced the term *cold hemagglutinin disease* to separate the disorder from other acquired hemolytic anemias [5]. With regard to the red blood cell (RBC) specificity of CAs, the serum of patients with CAs was said to contain "anti-I" ("I" for individuality) by Wiener et al. after testing a serum derived

from a patient against 22,964 RBC samples [6]. More recently, when methods of immuno-electrophoresis became available, the nature of CAs was identified as monoclonal M-type immunoglobulin (IgM) [7].

The terminology used for describing patients with AIHA is confusing [8]. The traditional classification of AIHA based on the results obtained with the DAT must be updated as follows (Table 1). Cold AIHAs have traditionally been classified as primary or "idiopathic" and secondary to viral infections or malignant diseases. However, nowadays, the primary or "idiopathic" form of cold agglutinin syndrome (CAS) is a well-defined clinicopathological entity and should be called a disease, not a syndrome.

**Table 1.** Classification of autoimmune hemolytic anemias (AIHA).

| Warm AIHA | Cold AIHA | Atypical AIHA |
|---|---|---|
| Idiopathic | Primary: cold agglutinin disease (CAD) | Warm and cold |
| Secondary | Secondary: cold agglutinin syndrome (CAS)Paroxysmal cold hemoglobinuria (PCH) | DAT-negative AIHA |

## 2. Definition of Cold Agglutinin Disease (CAD)

CAD is a designation used for a form of an acquired AIHA caused by a bone marrow lymphoproliferative disorder (LPD). The clonal B-cell lymphocytes produce monoclonal immunoglobulins (Ig), usually IgM-kappa that recognizes its own antigens located on the RBC membrane [9]. By far, the most common RBC antigen recognized by CAs is "I", a small minority recognize the "i" antigen and a few antibodies react with antigens other than I and i. The recognition of RBC antigens, particularly below core body temperature, provokes RBC agglutination in the acral circulation and acrocyanosis may appear. Extravascular hemolytic anemia is driven by classical pathway complement activation (Figure 1). Finally, thromboembolic risk is increased in CAD patients when compared with controls [10].

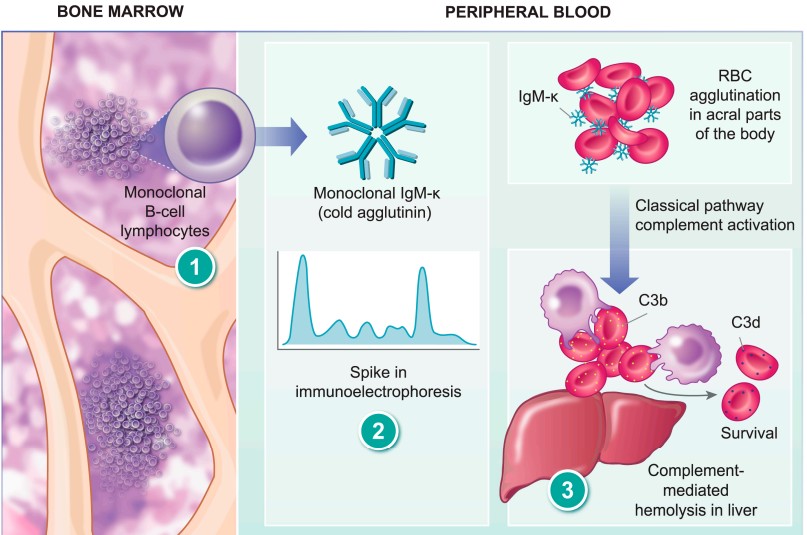

**Figure 1.** Cold agglutinin disease (CAD) is a form of acquired autoimmune hemolytic anemia (AIHA) caused by a bone marrow lymphoproliferative disorder (1). The clonal B-cell lymphocytes produce monoclonal immunoglobulin (Ig), usually IgM-kappa (2). IgM-κ acts as a cold autoantibody that recognizes its own red blood cells (RBC) producing RBC agglutination at low temperatures during passage through the acral parts of the circulation (3). Binding of IgM-κ to RBC antigens is a potent complement activator by the classical pathway from C1q to C3b. On rewarming to 37 °C in the central circulation, IgM-κ is detached and C3b-opsonized RBCs undergo phagocytosis in the liver, known as extravascular hemolysis. On the surviving cells, surface-bound C3b is degraded into C3d (3).

### 3. Epidemiology

It is usually said that cold AIHAs account for 15% of all AIHAs, the incidence rate of CAD is 1 case per million inhabitants per year, and the prevalence is 16 cases per million inhabitants. However, the majority of authors agree that these figures must be underestimated because the data come from small retrospective series [7,11]. A recent multinational, observational study was undertaken to improve knowledge of the epidemiological findings in CAD. Authors collected data from 232 patients in 5 countries (Norway, Italy, the U.K., Finland, and Denmark). Notwithstanding, they used data from Norway and Italy because the identification procedure of patients was considered to be population-based. For the first time, authors demonstrated a marked association of climate and incidence and prevalence of CAD. The incidence and prevalence of CAD was 1.9 cases/million per year and 20 cases/million, respectively, in Norway, with 0.48 cases/million per year and 5 cases/million, respectively, in Italy. They found a four-fold difference between cold and warm climates [12].

CAD affects middle-aged or elderly women, according to data obtained from an old and small-scale study [13]. This statement is confirmed by the previous population-based study in which authors found a male-to-female ratio of 0.56, with a mean age (range) at disease onset and at disease diagnosis of 67 years (32–94) and 68 years (33–96), respectively [12].

### 4. Pathogenesis of CAD

Cold reactive antibodies bind to the antigen at temperatures of 0–4 °C. CAs are cold autoantibodies that react with their own RBC antigens and produce RBC agglutination. The thermal amplitude of CAs is the highest temperature at which RBC agglutination occurs and CAs with thermal amplitude higher than 28–30 °C are pathogenic.

As stated before, nowadays we know that CAD is produced by a clonal LPD localized in the bone marrow that can be difficult to recognize and it remains a challenging diagnosis for pathologists [14]. For this reason, an expert central review of bone marrow biopsies is encouraged to increase the correct identification of this CA-associated LPD by hematopathologists [12]. Moreover, a centralized review offers promise for the clinician, the pathologist, and the patient [15].

The LPD found in the bone marrow in patients with CAD was previously classified into several entities of low-grade LPD, such as LPL or MZL [11]. A detailed histopathological study of 54 patients with CAD showed a homogenous lymphoid infiltration that has been termed CA-associated LPD [16]. The lymphoid infiltration usually consists of intraparenchymal nodular B-cell aggregates composed of small lymphoid cells and plasma cells. Infiltration can vary between 5% and 80% of the intertrabecular space. The immunophenotype of B cells shows CD20+, CD19+, CD22+, CD200+, IgMs+, and IgDs+. CD5 might be positive in less than half of the cases. CD10 and CD23 are negative. Immunoglobulin κ light chain is observed in 90% of cases. Mature plasma cells are seen surrounding the lymphoid nodular aggregates and throughout the marrow in between. The plasma cells have the same heavy and light chain restriction as the B-cells. The histological pattern does not display features typically found in LPL, such as paratrabecular infiltrates, fibrosis, lymphoplasmacytoid cell morphology, or infiltration by mast cells [17]. Bone marrow infiltration mimics that of MZL by morphology, immunophenotype, and molecular features [16]. However, CAD patients do not have an extramedullary MZL. These data suggest that CA-associated LPD, although exclusively present in the bone marrow, might be related to MZL. The transformation to large B-cell lymphoma is uncommon, probably occurring in less than 4% of the patients over 8 years [12].

Chromosome instability is one of the hallmarks of cancer. Malecka et al. studied 13 patients with CAD using cytogenetic microarrays and exome sequencing, and detected complete or partial gain of chromosome 3 (+3 or +3q) in all samples, barring one. Moreover, most cases showed either a gain of chromosome 12 or 18. These chromosome gains were mutually exclusive [18]. However, gains of chromosomes 3, 12, and 18 are not a specific

feature of CAD because they are also found in patients with MZL [19,20]. Similar to MZL [21], a gain of chromosomes 12 and 18 might be a predictor of therapy outcome [18].

The mutational landscape of CAD studied by whole exome sequencing and targeted sequencing showed mutations in genes known to be involved in lymphoma development [22]. Four genes showed nonsynonymous mutations in more than 20% of patients: *KMT2D* (67%), *IGLL5* (61%), *CARD11* (33%), and *CXCR4* (28%). All patients with either *CARD11* or *CXCR4* mutations, or both, had concurrent *KMT2D* mutations, and these patients presented lower hemoglobin levels at diagnosis compared to patients with an absence of the *KMT2D* mutation or patients with *KMT2D* mutations without *CARD11* or *CXCR4* mutations. Gene mutations observed in CAD affected the NF-KB pathway as well as chromatin modification or organization [22]. Importantly, the *MYD88* L265P mutation, present in over 90% of cases of lymphoplasmacytic lymphoma (LPL), was not found in any of the CAD cases. *CXCR4* mutations, described in up to 40% of LPL cases, tended to have a more aggressive disease at diagnosis [23].

Monoclonal B-cell lymphocytes produce monoclonal Ig, usually IgM-κ, and at molecular level, the *IGHV4-34* gene is the most frequent gene that encodes for the IgM heavy chain molecule found in CAD (more than 85%). Framework region 1 (FR1) of the heavy gene variable region is essential for recognition of the I antigen on the RBC membrane. It would appear that affinity and specificity for I antigen binding also depends upon the heavy chain complementarity determining region 3 (CDR3) and the light chain variable region [10]. The IG light chain is encoded by the *IGKV*3-20 and *IGKV*3-15 genes in more than 80% of patients, indicating that the light chain equally contributes to I antigenbinding [24].

Table 2 shows a summary of the main characteristics in order to differentiate among CA-associated LPD, LPL, and MZL.

**Table 2.** Differential diagnosis of CA-associated LPD, lymphoplasmacytic lymphoma (LPL), and marginal zone lymphoma (MZL).

| Characteristic | CA-Associated LPD | LPL | MZL |
|---|---|---|---|
| Histology | Intraparenchymal nodules | Interstitial, nodular, paratrabecular, and intrasinusoidal infiltrates | Intraparenchymal nodules and/or intrasinusoidal infiltrates (splenic) |
| Cytology | Small lymphoid cells, plasma cells | Small lymphocytes, lymphoplasmacytoid cells, and plasma cells | Small lymphocytes with abundant, pale cytoplasm and few admixed plasma cells |
| Cytogenetics | +3, +12, +18 | del(6q), gain(6p), +18 | +3, +12, +18 |
| *IGHV* gene | *IGHV*4-34 | *IGHV*3, *IGHV*3-23, *IGHV*3-7 | *IGHV*1-2 (splenic), *IGHV*3-4 (nodal) |
| *MYD88* L265P | Absent | Present (>90% cases) | Present (10% cases) |
| *KMT2D* | 67% | Absent | 34% |
| *CARD11* | 33% | Absent | 8% |
| *CXCR4* | 28% | 40% | Absent |
| Transformation to large B-cell lymphoma | Rare (3.4%) | Yes (5–13%) | Yes (15%) |

## 5. Clinical Presentation

CAD should be considered in an elderly patient with unexplained chronic hemolytic anemia, with or without cold-induced symptoms and/or thromboembolic complications.

### 5.1. Fatigue

Fatigue is usually considered a common symptom in patients with CAD and it is attributed to different factors, such as chronic anemia and underlying conditions. However, this symptom is poorly reported in the previous published literature. In a retrospective analysis of 89 patients at the Mayo Clinic, it was found that fatigue was reported at diagnosis and throughout the course of the disease in 21% and 40% of patients with CAD, respectively [7]. More detailed data about the level of fatigue can be found in a recent multicenter, open-label, single-group study that was conducted to assess the efficacy and safety of a new drug for patients with CAD and a recent history of blood transfusion [25]. The authors included 24 patients in the study, and a secondary end point was to assess

quality of life with the use of the Functional Assessment of Chronic Illness Therapy (FACIT) fatigue scale (scores range from 0 to 52, with a higher score indicating less fatigue). Patients with CAD had a mean baseline score on the FACIT fatigue scale of 32.5, a value similar to that reported in patients with advanced cancer or rheumatoid arthritis [25].

*5.2. Anemia*

Symptoms related to chronic anemia are not only very common in patients with CAD, but also vary greatly from patient to patient [13]. More than 90% of patients had anemia with a median (range) hemoglobin value of 92 g/L (45–153) [12]. Severe (hemoglobin <80 g/L), moderate (hemoglobin 80–100 g/L), and mild anemia (hemoglobin >100 g/L) was observed in 27%, 35%, and 38% of cases, respectively [11]. All series describing the clinical picture of patients with CAD include a number of cases with compensated hemolysis, thus, anemia is not detected [7,11,12]. This variability depends on the thermal range of the CAs as well as cold weather [13]. Nowadays, this assumption is challenged by a recent observational study, in which the authors have shown that hemolysis can persist throughout the year, although they confirmed that the variability of disease severity across patients is vast [26].

*5.3. Cold-Induced Symptoms*

As mentioned in the previous definition section, CAs recognize RBC antigens at low temperatures, which can be attained in the superficial skin vessels of acral parts of the body, such as the hands, feet, nose, and ears [13]. The result of RBC agglutination is pain, along with distal ischemia and the appearance of blue to deep purple in the extremities [27]. The thermal range of the CA is more important than the agglutination titer in explaining the severity of the symptoms. It is important to point out that skin manifestations in patients with CAD are the result of physical RBC agglutination and the use of Raynaud's phenomena to describe these symptoms are, strictly speaking, incorrect. Raynaud's disease is the consequence of vasoconstriction and Raynaud's phenomena are, in reality, three consecutive phenomena: first, the affected part becomes white and perhaps numb; secondly, it becomes swollen, stiff, and livid; and thirdly, vasoconstriction passes off and the part becomes red due to hyperemia [2]. Interestingly, the authors who published the largest series of patients with CAD classified them into 3 clinical phenotypes: type 1 (69% of cases) were patients with hemolytic anemia without cold-induced symptoms or only acrocyanosis; type 2 (21% of cases) were patients with hemolytic anemia with more severe acrocyanosis interfering with daily living or even gangrene or ulcerations; and type 3 (10% of cases) were patients with cold-induced symptoms and compensated hemolysis [12].

*5.4. Thromboembolic Complications*

An increased risk of thromboembolic events (TEs) in patients with many types of RBC hemolysis was noted in the past, although this was supported by low quality and a limited amount of data [13]. Nowadays, more data coming from three different studies are available. First, Ungprasert et al. conducted a systematic review and meta-analysis of four observational studies (three retrospective cohort studies and one cross-sectional study) comprising 13,036 patients with AIHA who had 472 venous thromboembolisms (VTE) [28]. The pooled risk ratio (RR) of VTE of patients with AIHA versus the control group was 2.63 (95% CI: 1.37 to 5.05). However, the statistical heterogeneity was high with an $I^2$ of 97%. To investigate this high heterogeneity, the authors performed a sensitivity analysis by excluding the cross-sectional study, which is generally considered a lower-quality design compared with cohort studies. After excluding this study, the $I^2$ was 0% and the pooled RR was 3.74 (95% CI: 3.39 to 4.13). Second, Bylsma et al. identified 72 patients with CAD between 1999 and 2013 in the Danish National Patient Registry and matched them to a general population comparison cohort of 720 individuals [29]. The risk of TEs was higher in the CAD patient cohort than in the comparison cohort: 1 year (7.2% vs. 1.9%), 3 years (9% vs. 5.3%), and 5 years (11.5% vs. 7.8%) after the index date. Third, Broome et al. identified

608 patients with CAD in the U.S.A. between 2006 and 2016 in the Optum Claims Clinical data set and matched them with 5873 patients without CAD [30]. Authors found that the overall risk of having a TE was 1.9 times higher in patients with CAD than in the patients without CAD during the study period. To summarize, although limitations exist in the previous data with regard to the previous three study designs, it seems clear that CAD is associated with an increased risk of TEs when compared with the general population [9].

## 6. Diagnosis

A direct antiglobulin test (DAT) must be ordered to demonstrate autoimmune pathogenesis when a hemolytic anemia is suspected in a patient, because of the presence of anemia with other signs of hemolysis (high LDH, high indirect bilirubin, high reticulocyte count, and low haptoglobin) [31]. It is important to note that the blood sample used for performing not only DAT, but also other laboratory parameters in a patient with suspected CAs should be kept warm (at 37–38 °C) after collection to avoid RBC agglutination. Prewarming at 37 °C for up to 2 h or a short preheating at 41 °C for 1 min may be tried to overcome the problems with the obtention of hemoglobin values and blood cell counts. Once plasma or serum is separated from the blood cells, the sample can be handled at room temperature [10,32].

### 6.1. Principles of DAT

DAT is a simple, quick, and inexpensive test with a good predictive value if it is performed when immune-hemolysis is suspected [33,34], and a blood sample is collected in a tube containing ethylenediaminetetraacetic acid (EDTA) as anticoagulant. In a physiological state, without hemolysis or anemia, it is known that RBCs can be opsonized with 100–500 molecules of IgG and/or 400–1000 molecules of complement (C3). However, under normal circumstances, RBCs contained in a sample collected in an EDTA tube do not autoagglutinate, due to the fact that their electrostatic charge causes them to mutually repel in solution (zeta potential). When RBC autoantibodies or C3 molecules are attached to RBC membrane (RBC sensitization), two main situations can occur. If RBCs are sensitized with IgG or C3, even with higher quantities than previously cited, they are incapable of overcoming the electrostatic repulsive force and antihuman globulin antisera (Coombs' reagent) is necessary to bridge the distance between RBCs sensitized with IgG and/or C3. In contrast, when RBCs are sensitized with as low as 50 molecules of IgM, this 1 million Daltons molecule is capable of spanning the intercellular distance between RBCs, and agglutination is visible to the naked eye after RBCs are incubated at low temperature (4 °C) [33].

### 6.2. Methods of Performing DAT

As with other basic procedures in immunohematology, the DAT principle is to detect an antigen–antibody reaction. Traditionally, the evidence of the formation of this reaction in vitro has been the visualization of agglutinates or the presence of hemolysis within the tube test. Nowadays, manual tube tests have been replaced by automated systems, such as the column agglutination method. The sensitivity limit of the routine tube method or the column agglutination method is almost 200 and 100 IgG molecules/RBC, respectively [35]. Although automation within a laboratory has a considerable number of advantages, when DAT is performed with this new technology, one must have in mind that the final accuracy of the test is different when compared with old, manual tube agglutination tests. Barcellini et al. compared the DAT results obtained with these two methods and they observed a sensitivity and specificity of tube test of 43% and 87%, respectively. The sensitivity and specificity of the microcolumn test was 70% and 65%, respectively [36]. The authors confirmed that the DAT tube test was the gold standard in diagnosing AIHA, although they also found that some negative results with the DAT tube test were identified with microcolumn test, reflecting the different accuracy of these laboratory tests to detect the low quantity of IgG bound on the RBC membrane [37]. More sensitive techniques can be used

to detect less than 100 IgG molecules/RBC, such as flow cytometry [38] or immunoradiometric assays. In one study using radioimmunoassay, the lowest limit to detect RBC-bound IgG was 78.5 IgG molecules/RBC [39].

*6.3. Results of DAT in CAD and Further Characterization*

CAs are suspected after obtaining a positive DAT for C3. The next steps must be undertaken to define thermal amplitude, specificity and titer of CAs (Table 3). Thermal amplitude is the highest temperature at which the CA will bind to its antigen. Specificity is the antigenic determinant recognized by the CA. The CA titer is the inverse value of the highest serum dilution at which agglutination can be detected.

**Table 3.** Immunohematologic study when CAD is suspected.

| Study | Results |
|---|---|
| Direct antiglobulin test | Positive for C3 (negative or weakly positive for IgG) |
| Titration | $\geq 64$ |
| Specificity | I, i |
| Thermal amplitude | $>4\,^{\circ}\text{C}$ |

After the characterization of CAs, serum monoclonal IgM-κ with electrophoresis and immunofixation is detected in more than 90% of patients [40]. Only 7% of cases show λ light chain restriction and IgG class is observed in less than 5% of cases [10,11].

Finally, a bone marrow biopsy is mandatory to discover the specific bone marrow histopathological pattern previously detailed and termed as "CA-associated LPD".

## 7. Treatment

All treatment should aim at an improved quality of life. Therefore, each patient must be carefully assessed in order to guide therapy decisions depending upon his/her main complaint. As stated before, fatigue, anemia, cold-induced symptoms, and TE complications are common in patients with CAD. If patients have no relevant symptoms or problems, a "watch and wait" decision is appropriate. General measures, such as folic acid and B12 vitamin supplementation, avoiding low temperatures, and early treatment of bacterial infections can be used to manage these patients. When a patient needs a more specific therapeutic approach, physicians should consider the availability of clinical trials at all stages of treatment.

Glucocorticoids, other active therapies in patients with warm AIHA, and splenectomy must not be used to treat patients with CAD because response rates are low and the liver is the dominant site of destruction of C3-sensitized RBCs [41]. Moreover, patients who respond to steroids need a high dose of medication to maintain remission with an unacceptable number of adverse events [9,42].

*7.1. Fatigue and Anemia*

Thanks to a clinical trial performed in patients with CAD and a recent history of blood transfusion, we are now aware that a clinically meaningful reduction in fatigue occurred very fast (within one week) after starting treatment with sutimlimab [25]. Sutimlimab is a humanized monoclonal antibody that selectively targets the C1s protein. By acting at this juncture, classical pathway complement activation is stopped. The authors hypothesized that fatigue, in addition to anemia, could be related to the inactivation of complement pathway or hemolytic activity, or both. However, as authors stated, the open-label design of the study makes the interpretation of these results difficult. Another study with sutimlimab administered to patients who suffered CAD without a recent history of blood transfusion has just finished, and the results are keenly awaited (Clinicaltrials.gov number NCT03347422).

## 7.2. Cold-Induced Symptoms and TE Complications

Acrocyanosis is common in patients with CAD, but it is rarely an indication for starting treatment. However, descriptive studies have shown that up to 80% of patients have received pharmacological treatment [7,11]. Thermal protection is enough to manage this symptom, and only when severe cold-induced symptoms appear is it necessary to perform other therapeutic approaches with the aim of removing IgM from the circulation [27]. New treatments aimed at inhibiting classical pathway complement activation are not useful for treating acrocyanosis, because this clinical manifestation is due to the binding of Cas to RBC antigen. With regard to TE complications, it is unclear whether treatment of the hemolytic process itself could reduce the risk of VTE. Thus, thromboprophylaxis is recommended in acute exacerbations or for chronic disease in risk situations [40].

## 7.3. Emergency Situations

When an acute trigger develops a severe or life-threatening anemia, urgent therapy is necessary to remove Cas from circulation [27]. Plasma exchange can be an excellent approach, given the fact that most of the IgM is located in intravascular space and plasma exchange is highly efficient in removing intravascular molecules [43]. In fact, severe CAD is considered a category II (second line treatment) indication for performing plasma exchange in the evidence-based guidelines published by the American Society for Apheresis (ASFA) [44]. However, this apheresis procedure has to be considered part of a whole treatment strategy consisting in removing IgM with plasma exchange and blocking the production of new CAs with an associated treatment, such as rituximab. This approach was used by our group for treating patients with not only severe AIHAs [45,46], but also with different severe autoimmune diseases [47–49]. A meta-analysis published in 2020 found that the use of plasma exchange may yield a 22% increase in the incidence of AIHA remission compared to no plasma exchange, and that plasma exchange may also increase the incidence of adverse events by 12% compared to the control group, although this increase was not statistically significant [50]. The authors concluded that plasma exchange may be beneficial for short-term control of AIHA. However, it should be noted that the authors could not establish the efficacy of plasma exchange in warm and cold AIHA subgroups, due to a lack of subgroup data.

## 7.4. Current Recommendations

A treatment directed to deplete the B-cell clone located in the bone marrow seems to be a reasonable approach to treat patient with CAD [9]. There are no prospective, randomized trials in this setting, but different prospective, non-randomized, and "real life" observational studies support the use of different chemoimmunotherapy regimens. Table 4 summarizes that rituximab, alone or in combination with bendamustine, shows a beneficial effect with low toxicity.

**Table 4.** Current recommended treatments for patients with CAD.

| Reference | Therapy | n | Overall Response | Response Duration | Toxicity |
|---|---|---|---|---|---|
| [51] | Rituximab | 37 | 54% | 11 months | Low |
| [52] | Rituximab+bendamustine | 45 | 71% | >32 months | Low |
| [53] | Rituximab+fludarabine | 29 | 76% | >66 months | Significant |
| [54] | Bortezomib | 19 | 32% | 216 months | Low |

When these previous treatments are not effective, a small series suggests that ibrutinib, a Bruton's tyrosine kinase inhibitor, could be effective [55]. Finally, new drugs under development, such as sutimlimab [25], as previously cited, and pegcetacoplan [41], are directed to inhibit the classical pathway complement activation.

## 8. Conclusions

CAD is an acquired form of AIHA caused by a distinct B-cell LPD in the bone marrow named CA-associated LPD. Monoclonal B-cell lymphocytes produce monoclonal IgMκ, with the ability to bind to I antigen on the RBC membrane at low temperatures. Restrictions, not only in *IGHV* but also in immunoglobulin light chain *V*, contribute to I antigen-binding on the RBC membrane. RBCs agglutinate in the acral parts of the body causing acrocyanosis, while the classical pathway complement activation by CAs is responsible for extravascular hemolysis in the liver of C3b-sensitized RBCs. The accurate diagnosis of CA-associated LPD is a challenge for pathologists and hematologists, and it highlights the importance of integrating clinical, analytical, and pathological data.

**Author Contributions:** Conceptualization, F.C., J.C. and A.S.; Writing—original draft preparation, F.C. and J.C.; Writing—review and editing, A.S. All authors have read and agreed to the published version of the manuscript.

**Funding:** This research received no external funding.

**Acknowledgments:** Authors acknowledge Alison Schroeer (www.illustratingscience.com; last accessed date 11 February 2022) for drawing Figure 1. This text was revised by David Kennedy (MCIL), Member of the United Kingdom's Chartered Institute of Linguists.

**Conflicts of Interest:** The authors declare no conflict of interest.

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
