# Peer review of "Cold Agglutinin Disease: A Distinct Clonal B-Cell Lymphoproliferative Disorder of the Bone Marrow"

_hemato, doi:10.3390/hemato3010014_

Round 1

Reviewer 1 Report

This is a very nicely written paper which captures the history and testing of CAD very well.

Personally I enjoyed reading the history, and I felt the whole paper was balanced and non-biased.

I would add a bit more about sample handling as this is a huge gap in clinical practice and is often badly managed in clinical settings- this will compliment the lab section very nicely.

The treatment section is quite short- but I feel that the key information is present- the R/ chemo data are reflective of the reality and fairly presented.

The sutimlimab data are well cited- this is work-in-progress.

Author Response

Reviewer 1.

This is a very nicely written paper which captures the history and testing of CAD very well.

Personally I enjoyed reading the history, and I felt the whole paper was balanced and non-biased.

I would add a bit more about sample handling as this is a huge gap in clinical practice and is often badly managed in clinical settings- this will compliment the lab section very nicely.

The treatment section is quite short- but I feel that the key information is present- the R/ chemo data are reflective of the reality and fairly presented.

The sutimlimab data are well cited- this is work-in-progress.

Authors’ answers:

Authors would like to thank the reviewer for those kind words. Authors added a comment regarding sample handling in lines 241-247.

Reviewer 2 Report

 Clement et al. provide an excellent review of Cold agglutin disease, an entity which probably is highly underdiagnosed.  The following are some minor suggested changes.

  1. In abstract change clinical pictures to clinical picture (line 16)
  2. Wait and watch is a more common phraseology than wait and see (line 17 abstract)
  3. Change “Classical classification” to “traditional classification” (line 51)
  4. Expand CAS (cold agglutin syndrome) the first time this terminology is introduced in line 54
  5. C3d is show in the figure but is not references anywhere in the figure legend or the text.
  6. Please rephrase this sentence “However, the majority of authors agree that these figures must be 80 underestimated because CAD is a rare disease with low incidence and prevalence and 81 data come from small retrospective series”. It’s a bit unclear, the low numbers being a result of underestimation is not supported by the statement that it’s a rare disease.
  7. Please restructure as” Finally, new drugs under development such as sutimlimab, as previously cited, and pegcetacoplan,  are directed to inhibit the classical pathway complement activation. (line 350-352)
  8. In Table 2, please specify that amongst MZL, splenic MZL has intrasinusoidal infiltrates.
  9. Please expand DAT the first time it is used, line 231
  10. Please explain “proper tube”, are the authors referring to the specific anti-coagulant used in the collection tube e. EDTA? (line 238)
  11. Is anything know in regard to the flow phenotypic findings for these LPDs, can they be CD5 or CD10 positive, do we have information on the CD20 and light chain intensity as well as CD200? Do we know about the CD19 and CD56 expression on the plasma cells?
  12. HE picture and CD20 IHC of the bone marrow lymphoid aggregates might be of interest to most pathologists. Would the authors be able to provide that?

Author Response

Reviewer 2.

Clement et al. provide an excellent review of Cold agglutin disease, an entity which probably is highly underdiagnosed.  The following are some minor suggested changes.

  1. In abstract change clinical pictures to clinical picture (line 16)

Done.

  1. Wait and watch is a more common phraseology than wait and see (line 17 abstract)

Done.

  1. Change “Classical classification” to “traditional classification” (line 51)

Done.

  1. Expand CAS (cold agglutin syndrome) the first time this terminology is introduced in line 54

Done.

  1. C3d is show in the figure but is not references anywhere in the figure legend or the text.

Authors explained C3d in figure legend.

  1. Please rephrase this sentence “However, the majority of authors agree that these figures must be 80 underestimated because CAD is a rare disease with low incidence and prevalence and 81 data come from small retrospective series”. It’s a bit unclear, the low numbers being a result of underestimation is not supported by the statement that it’s a rare disease.

Authors rephrased the sentence.

  1. Please restructure as” Finally, new drugs under development such as sutimlimab, as previously cited, and pegcetacoplan,  are directed to inhibit the classical pathway complement activation. (line 350-352)

Done.

  1. In Table 2, please specify that amongst MZL, splenic MZL has intrasinusoidal infiltrates.

Done.

  1. Please expand DAT the first time it is used, line 231

Done.

  1. Please explain “proper tube”, are the authors referring to the specific anti-coagulant used in the collection tube e. EDTA? (line 238)

Done.

  1. Is anything know in regard to the flow phenotypic findings for these LPDs, can they be CD5 or CD10 positive, do we have information on the CD20 and light chain intensity as well as CD200? Do we know about the CD19 and CD56 expression on the plasma cells?

Authors added phenotypic findings in lines 116-119.

  1. HE picture and CD20 IHC of the bone marrow lymphoid aggregates might be of interest to most pathologists. Would the authors be able to provide that?

Authors agree with the reviewer. However, we don’t have a high quality image to be published.